# *let-7* coordinates the transition to adulthood through a single primary and four secondary targets

Florian Aeschimann[1,2] , Anca Neagu[1] , Magdalene Rausch[1], Helge Großhans[1,2]

The juvenile-to-adult (J/A) transition, or puberty, is a period of extensive changes of animal body morphology and function. The onset of puberty is genetically controlled, and the *let-7* miRNA temporally regulates J/A transition events in nematodes and mammals. Here, we uncover the targets and downstream pathways through which *Caenorhabditis elegans let-7* controls male and female sexual organ morphogenesis and skin progenitor cell fates. We find that *let-7* directs all three processes by silencing a single target, the post-transcriptional regulator *lin-41*. In turn, the RNA-binding protein LIN41/TRIM71 regulates these processes by silencing only four target mRNAs. Thus, by silencing LIN41, *let-7* activates LIN-29a and MAB-10 (an early growth response-type transcription factor and its NAB1/2-orthologous cofactor, respectively) to terminate progenitor cell self-renewal and to promote vulval integrity. By contrast, *let-7* promotes development of the male sexual organ by up-regulating DMD-3 and MAB-3, two Doublesex/MAB-3 domain–containing transcription factors. Our results provide mechanistic insight into how a linear chain of post-transcriptional regulators diverges in the control of a small set of transcriptional regulators to achieve a coordinated J/A transition.

## Introduction

Development of multicellular organisms requires faithful control of cell fates in both space and time. Although the mechanistic understanding of temporal control lags behind that of spatial patterning, fundamental and conserved regulators of developmental timing in animals have been identified. In particular, the *let-7* miRNA and its regulator LIN28 control stem or progenitor cell fates and the timing of sexual maturation of organisms as distinct as worms and mammals (Corre et al, 2016; Faunes & Larrain, 2016).

Both *let-7* and *lin-28* were initially described in *Caenorhabditis elegans* (Ambros & Horvitz, 1984; Moss et al, 1997; Reinhart et al, 2000), where a highly stereotypic development has enabled the identification of molecular factors that control developmental

timing. Genetic screens have uncovered "heterochronic" mutations that change the timing of specific developmental events relative to others (Ambros & Horvitz, 1984). Along with several other genes, *let-7* and *lin-28* form a heterochronic pathway that controls the transition from a juvenile to an adult animal (Ambros, 1989; reviewed in Rougvie & Moss (2013)). This transition involves formation of mature sexual organs, that is, morphogenesis of the vulval-uterine tract in hermaphrodites (Ecsedi et al, 2015) and cell retraction events that shape the male tail (Del Rio-Albrechtsen et al, 2006). Hermaphrodites with dysfunctional *let-7* rupture through the vulva and die (Reinhart et al, 2000). In addition, *let-7* controls the fate of epidermal progenitor cells at the juvenile-to-adult (J/A) transition. These so-called seam cells divide asymmetrically once in each larval stage, but exit the cell cycle during transition into adulthood (Sulston & Horvitz, 1977). In *let-7*–mutant animals, seam cells do not switch to the adult fate and continue to self-renew (Reinhart et al, 2000), consistent with a conserved function of *let-7* in controlling stem and progenitor cell fates in mammals (reviewed in Büssing et al (2008)).

Although the *let-7* miRNA exhibits perfect sequence conservation from worm to human and is the only heterochronic gene essential for larval survival, the molecular basis of *let-7*–mutant phenotypes has only begun to emerge. miRNAs silence target mRNAs post-transcriptionally by binding short stretches of complementary sequence, typically located in the 3′ UTR. Sequence complementarity of as little as seven nucleotides can be sufficient to induce silencing, and sequences complementary to *let-7* occur in numerous mRNAs. Accordingly, many targets of *let-7* have been reported (Abrahante et al, 2003; Lin et al, 2003; Großhans et al, 2005; Andachi, 2008; Jovanovic et al, 2010; Hunter et al, 2013; ). However, it has been unclear which and how many of these targets are functionally important. To resolve this issue, we have recently developed a genetic approach to uncouple endogenous *let-7* targets from silencing. Thus, we showed that *lin-41* is the only target that *let-7* needs to silence to prevent vulval bursting and death (Ecsedi et al, 2015).

Here, we report that also the other major phenotypes of *let-7*–mutant animals are caused by failure to silence the single target LIN41 at the J/A transition. (We will refer to the gene by its worm-specific hyphenated name, *lin-41*, and to the protein by its generic

[1]Friedrich Miescher Institute for Biomedical Research, Basel, Switzerland   [2]University of Basel, Basel, Switzerland

Correspondence: helge.grosshans@fmi.ch

name, LIN41, to reflect phylogenetic conservation.) To understand how the down-regulation of LIN41 triggers the transition to adulthood, we characterized LIN41 targets. Although LIN41 proteins can function as E3 ubiquitin ligases and structural proteins (reviewed in Ecsedi & Großhans (2013)), we had previously demonstrated that *C. elegans* LIN41 binds and post-transcriptionally silences four transcripts (Aeschimann et al, 2017). Here, we show that LIN41-mediated silencing of these four mRNAs, which encode the transcription factors LIN-29a, DMD-3, and MAB-3, and the transcription cofactor MAB-10, explain *let-7*–mutant phenotypes. The four transcriptional regulators act in two pairs in different tissues. Thus, through repression of LIN41, *let-7* promotes adult seam cell fates and vulval integrity by activating *lin-29a* and *mab-10*, and cell retraction events during male tail morphogenesis by activating *mab-3* and *dmd-3*. A partially redundant activity of LIN41 targets combines with isoform-specific and spatially restricted silencing of *lin-29a* to explain why previously described phenotypes overlapped only partially between LIN41 target gene and *let-7*–mutant animals (Hodgkin, 1983; Euling et al, 1999; Del Rio-Albrechtsen et al, 2006; Mason et al, 2008; Ecsedi et al, 2015). Thus, our results extend the mechanistic and conceptual understanding of a paradigmatic temporal patterning pathway. They identify *let-7*–LIN41 as a versatile regulatory module and reveal how several levels of post-transcriptional regulation promote transition into adulthood through controlling transcriptional programs.

## Results

### LIN41 is the single key target of *let-7* for three distinct developmental functions

We previously showed that regulation of LIN41 alone accounted for the function of *let-7* in *C. elegans* vulval development (Ecsedi et al, 2015). Given that numerous targets of *let-7* had been reported, we asked whether additional known phenotypic functions of *let-7* involved other targets than LIN41. To explore this, we analyzed a set of three different mutants that allowed us to test whether a certain phenotype is caused by failed *let-7*–mediated repression of *lin-41* only (Fig 1A) (Ecsedi et al, 2015). First, we studied *let-7*–mutant phenotypes using *let-7(n2853)* animals. These worms harbor a G-to-A point mutation in the *let-7* seed sequence that abrogates *let-7* activity at 25°C (Reinhart et al, 2000) and thereby causes up-regulation of all *let-7* targets, including *lin-41*. We will refer to this allele as *let-7(PM)* for *let-7* point mutation. Second, we tested if preventing *let-7*–mediated silencing of only *lin-41* is sufficient to recapitulate a phenotype. To this end, we used *lin-41(xe8)*–mutant animals that lack a segment of the *lin-41* 3′ UTR that contains the two *let-7* target sites. We refer to *lin-41(xe8)* in the following as *lin-41(ΔLCS)*, where LCS stands for *let-7* complementary sites (Vella et al, 2004). Third, we examined if de-repression of *lin-41* was necessary for phenotypes in *let-7*–mutant animals, by asking whether restoring *let-7*–mediated silencing of *lin-41* but not of the other targets sufficed to suppress a given phenotype. We used *lin-41(xe11); let-7(n2853)* double mutant animals, in which both *let-7* target sites on the *lin-41* 3′ UTR contain a compensatory point

mutation (*CPM*) to allow base pairing to the *let-7(PM)*–mutant miRNA (Fig 1A). This mutant combination results in substantial, albeit incomplete, repression of *lin-41*, whereas the other *let-7* target mRNAs are de-silenced to the same extent as in *let-7(PM)* single mutant animals (Fig 1A) (Ecsedi et al, 2015; Aeschimann et al, 2017). For completeness, we also studied the phenotypes of *lin-41(xe11)* single mutant animals, for which *lin-41* silencing is also incomplete because binding of wild-type *let-7* to the mutant LCS's is impaired (Ecsedi et al, 2015; Aeschimann et al, 2017). We refer to *lin-41(xe11)* as *lin-41(CPM)*.

This set of mutant strains allowed us to test which of the developmental events triggered by *let-7* are transmitted by *lin-41* and which events involve other targets (Fig 1B). Specifically, in addition to the role of *let-7* in vulva morphogenesis, we explored its functions in the control of seam cell self-renewal and male tail retraction. Consistent with previous observations (Ecsedi et al, 2015), the failure to down-regulate *lin-41* explained the lethality of the *let-7* mutation, as this phenotype was recapitulated in *lin-41(ΔLCS)* animals and rescued in *lin-41(CPM); let-7(PM)* animals (Fig 1C and Table S1). A vulval bursting phenotype was also virtually absent from *lin-41(CPM)* single mutant animals (Table S1).

We next examined *let-7*–mediated control of seam cell proliferation. Loss of *let-7* activity causes a failure of seam cells to exit the cell cycle (Reinhart et al, 2000), resulting in additional seam cell divisions at the young adult stage. We quantified this phenotype by counting the number of GFP-marked seam cell nuclei at the late L4 stage and a few hours later, in young adulthood, shortly before *let-7 (PM)* and *lin-41(ΔLCS)* animals died. At the L4 stage, just before the final molt, both *lin-41(ΔLCS)*– and *let-7(PM)*–mutant animals exhibited a wild-type number of seam cells (Fig 1E, gray circles; Table S2). However, after the molt, both strains of mutant animals had more seam cells than wild-type animals, reflecting a failure in termination of the self-renewal program at the transition to adulthood (Fig 1E, black circles; Table S2). Although a partial desilencing of *lin-41* in *lin-41(CPM)*–mutant animals caused no extra seam cell divisions (Table S2), a complete uncoupling of *lin-41* from *let-7* in *lin-41(ΔLCS)*–mutant animals recapitulated the *let-7(PM)* phenotype both qualitatively and quantitatively (Fig 1D and E and Table S2). Hence, we conclude that dysregulation of *lin-41* is sufficient for this *let-7*–mutant phenotype. Moreover, because restored repression of *lin-41* in *lin-41(CPM); let-7(PM)* double mutant animals sufficed to revert seam cell numbers to the lower, wild-type level (Fig 1D and E and Table S2), *lin-41* dysregulation is also necessary for the phenotype. Thus, LIN41 is the single target of *let-7* for a second function, that is, control of seam cell self-renewal.

Finally, we analyzed the tails of adult males. In wild-type animals, cells in the male tail undergo retraction to form the mature male reproductive organ (Nguyen et al, 1999). Mail tail cell retractions were previously found to be delayed in *let-7(PM)* as well as in *lin-41* gain-of-function (gf) males (Del Rio-Albrechtsen et al, 2006). However, *let-7(PM)* mutants were analyzed only at 15°C, a semi-permissive temperature for survival and presumably other phenotypes. Moreover, the *lin-41gf* alleles used were considered weak, and the mechanistic basis of their hyperactivation, involving mutations in the first coding exon of *lin-41*, is unknown. Strikingly, *let-7 (PM)* animals grown at 25°C and *lin-41(ΔLCS)*–mutant animals exhibited phenotypes that were much more severe than those

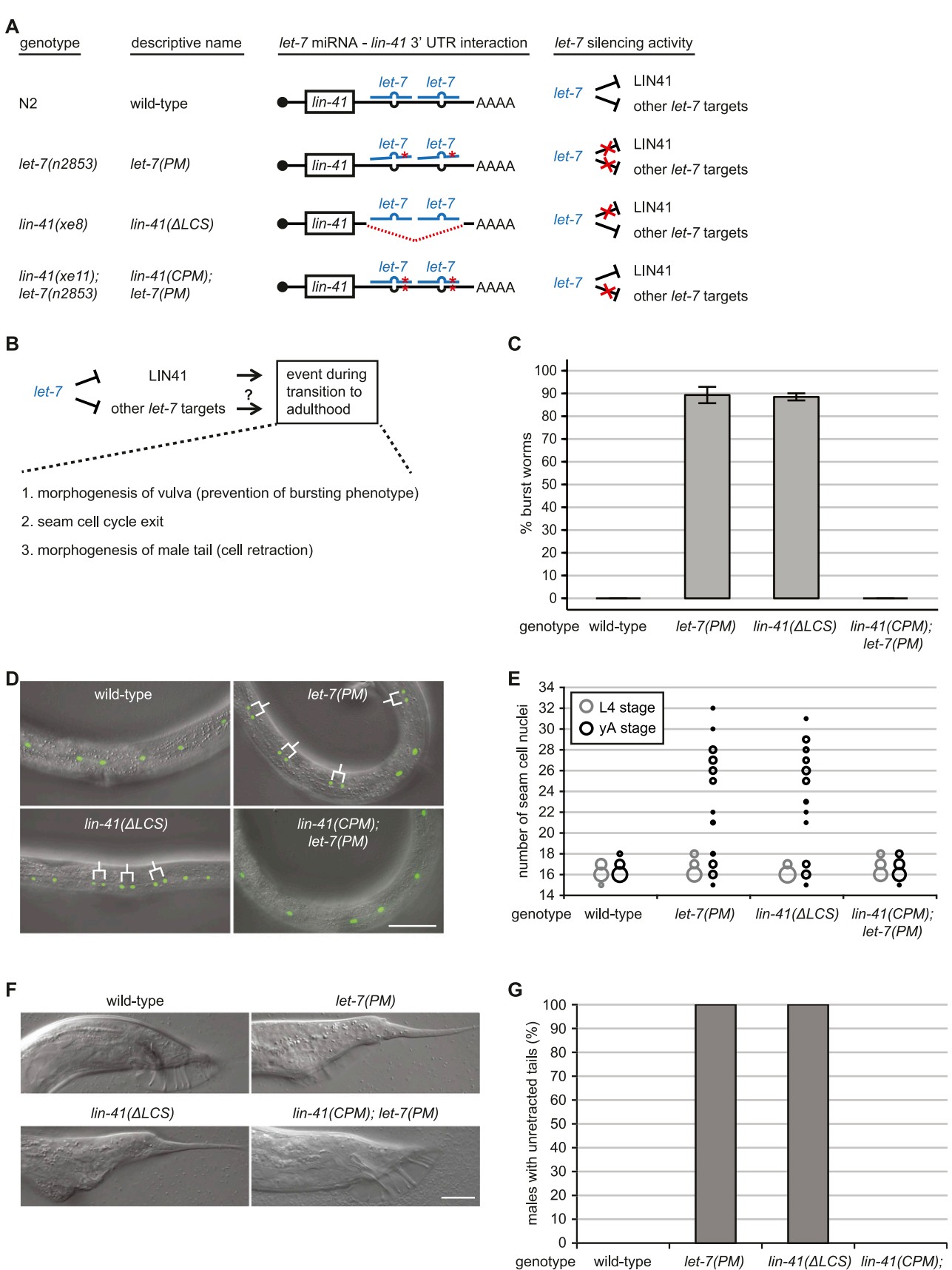

caused by a mere delay in tail cell retractions. As analyzed in more detail in the following section, almost all *let-7(PM)* and all *lin-41(ΔLCS)*–mutant adult males had "unretracted" tails with a long, pointed shape, resembling those of hermaphrodites (Fig 1F and G and Table S3). By contrast, the male tails of *lin-41(CPM); let-7(PM)* double mutant animals were of normal, wild-type appearance (Fig 1F and G and Table S3). Hence, our data suggest that *let-7* triggers male tail development through the single target *lin-41*. We conclude that all three *let-7*–mutant phenotypes analyzed here can be fully explained by dysregulation of *lin-41* and that *lin-41* is the key target of *let-7* for different developmental functions and in different tissues.

### Sustained LIN41 expression leads to a complete failure of male tail cell retraction

Because the mail tail retraction phenotype of *lin-41(ΔLCS)* and *let-7 (PM)*–mutant animals were much more severe than what was reported earlier for *lin-41gf*–mutant animals and *let-7(PM)*–mutant animals grown at 15°C (Del Rio-Albrechtsen et al, 2006), we investigated this phenotype in more detail.

During the L4 stage, male tails are reshaped from a pointed into a more rounded structure, as a consequence of cells moving anteriorly (Nguyen et al, 1999). In wild-type males, the first visible step of this process occurs at the mid L4 stage, when the four epidermal cells hyp8-11 at the very tip of the tail fuse and start to retract anteriorly, withdrawing from the larval cuticle and turning the tip into a rounded shape (Fig 2A(ii), dashed arrow). At the late L4 stage, the entire tail retracts, leaving behind male-specific sensilla called rays (Fig 2A(iii), arrow). After the last molt, the adult male tail is freed from the larval cuticle, but keeps a structure called the fan (Fig 2A(iv), arrowhead), consisting of a fold in the outer layer of the adult cuticle.

Del Rio-Albrechtsen et al (2006) previously characterized the phenotypes of *lin-41(bx37)* and *lin-41(bx42) gf* mutations on male tail development. Both weak *gf* alleles caused a "leptoderan" (Lep) phenotype, which we confirmed (Figs 2B(iv) and S1B and Table S3). The Lep phenotype is characterized by a selectively delayed retraction of hyp8-11 but not of the other cells, giving rise to animals with tails that appear wild-type in shape except for the presence of a tail spike (Nguyen et al, 1999). We observed a similar, but less penetrant phenotype with *lin-41(CPM)* (Figs 2B and S1B and Table S3). By contrast, rather than being merely delayed, tail tip cell retraction did not occur at all in *lin-41(ΔLCS)* and in *let-7(PM)* males (Fig 2A).

To compare the different mutants in more detail, we quantified tail cell retraction phenotypes during the late L4 stage (Fig 2C and Table S3) and final tail phenotypes in young adults (Fig S1B and

Table S3). Most *lin-41(bx37)–* and *lin-41(bx42)*–mutant males had a partially retracted tail at the late L4 stage, with a spike protruding from the otherwise normal tail (Fig 2B(iii) and C and Table S3). By contrast, none of the *lin-41(ΔLCS)*–mutant males displayed any cell retraction (Fig 2A(iii) and C and Table S3). At the young adult stage, this resulted in Lep tails for *lin-41(bx37)* or *lin-41(bx42)* mutants, whereas *lin-41(ΔLCS)* and *let-7(PM)* animals had a distinct and more severe "unretracted" phenotype (Figs 2A(iv) and B(iv), and S1B; Table S3). Despite the severity of the defect observed with *let-7(PM)* males grown at 25°C, male tail development was fully restored in *lin-41(CPM); let-7(PM)* double mutant animals (Fig 2A and C and Table S3). From this analysis, we conclude that proper male tail development absolutely depends on repression of *lin-41* by *let-7*. We propose that the previously reported Lep phenotypes of *lin-41gf–* and *let-7(PM)*–mutant animals grown at 15°C reflects the fact that, under these conditions, *let-7* will still eventually silence *lin-41*, but with a delay and/or less extensively. By contrast, *let-7*–mediated silencing is altogether lost in *let-7(PM)*–mutant animals at 25°C and in *lin-41(ΔLCS)* animals. We further note that male tail morphogenesis appears particularly sensitive to LIN41 activity levels, since the weak *gf* alleles *lin-41(bx37)*, *lin-41(bx42)*, and *lin-41 (CPM)* cause delayed, albeit largely functional male tail retraction (Fig 2B), but no seam cell division or survival phenotypes (Fig 1 and Tables S1, S2, S3, and Del Rio-Albrechtsen et al, 2006).

### LIN41 specifically binds to only a few somatic mRNAs

We previously identified four mRNA targets of LIN41 with ribosome profiling experiments that revealed gene expression changes in mutant animals with dysregulated LIN41 expression (Aeschimann et al, 2017). Whether and to what extent these targets mediate any or all of the LIN41 functions in the transition to adulthood and whether additional functionally relevant targets exist have remained untested. To start addressing these questions, we sought to identify targets of LIN41 globally in the developmental period during which *let-7* initiates repression of LIN41. Therefore, we used RNA co-immunoprecipitation coupled to RNA sequencing (RIP-seq) on a mixture of L3- and L4-stage animals. We used an anti-FLAG antibody to enrich for mRNAs bound by LIN41, which we expressed from a rescuing *flag::gfp::lin-41* transgene in the *lin-41(n2914)*–mutant background (Aeschimann et al, 2017). Wild-type worms expressing *flag::gfp::sart-3* (Rüegger et al, 2015) served as a negative control. To identify candidate targets in both sexes, we performed RIP-seq in a *him-5(e1490)* genetic background, which increases the incidence of males in a population (~35% males versus <1% in a wild-type population; Meneely et al, 2012).

---

**Figure 1. Failure of LIN41 down-regulation explains multiple *let-7*–mutant phenotypes.**
**(A)** Schematic illustration of *let-7–* and *lin-41* 3′ UTR–mutant alleles (not to scale) and of the *let-7* silencing activities in the different mutant backgrounds. Red asterisks depict point mutations and the red dotted line indicates a deletion. For both *let-7* target sites on the *lin-41* 3′ UTR in the *lin-41(xe11); let-7(n2853)* background, a wild-type G:C base pair is replaced by an A:U base pair. This rescues *lin-41* down-regulation by *let-7*, although not to the full extent (Aeschimann et al, 2017). **(B)** Schematic of a section of the heterochronic pathway regulating the onset of events during the J/A transition. The experiments of Fig 1 test if these events are regulated by silencing of only one *let-7* target (LIN41) or by silencing of any other combination of *let-7* targets. **(C)** The percentage of burst adult worms of the indicated genotypes grown in synchronized populations at 25°C for 45 h. Data as mean ± SEM of N = 3 independent biological replicates with n ≥ 400 worms per genotype and replicate. **(D)** Example micrographs of young adult worms expressing transgenic *scm::gfp* to visualize seam cell nuclei. Branched lines indicate seam cells originating from extra cell divisions. Scale bar: 50 µm. **(E)** Quantification of seam cell nuclei at the late L4 larval (L4) or young adult (yA) stage, in worms of the indicated genetic backgrounds. Areas of bubbles represent the percentage of worms with the corresponding number of seam cells. n = 20 for L4, n > 50 for yA, worms grown at 25°C. **(F)** Example micrographs of tails in adult males of the indicated genetic background. Scale bar: 20 µm. **(G)** The percentage of young adult males of the indicated genotype with unretracted tails. n ≥ 100, worms grown at 25°C.

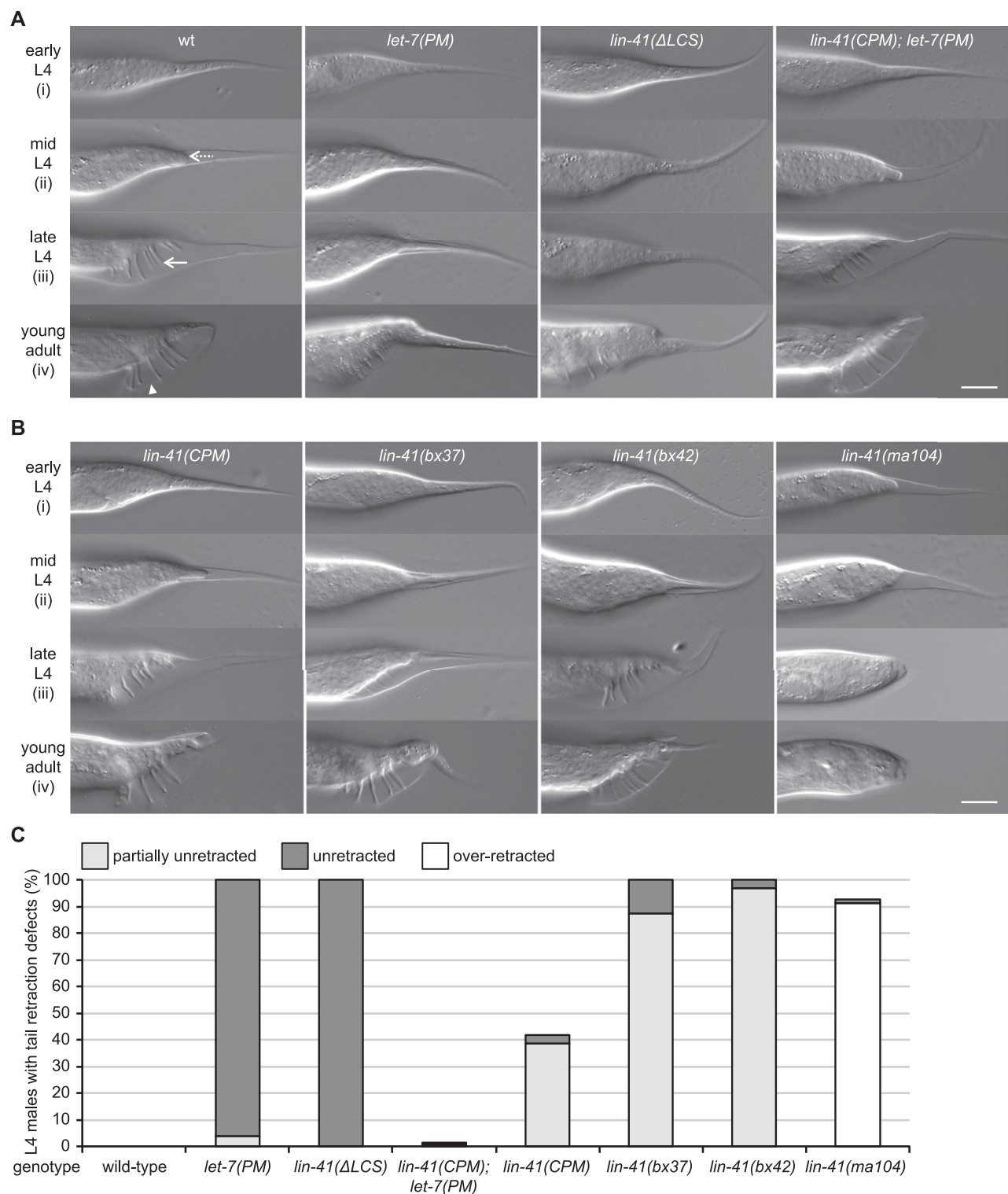

**Figure 2. A failure in LIN41 down-regulation results in a complete loss of male tail cell retractions.**
**(A, B)** Example micrographs of male tails at different developmental time points in the indicated genetic backgrounds. The dashed arrow illustrates the anterior retraction of the epidermal cell at the tail tip. The full arrow and the arrowhead point to one of the rays and the fan, respectively. Scale bars: 20 μm. **(C)** Quantification of the male tail phenotypes of the indicated genotypes at the late L4 larval stage as illustrated with pictures (iii) in (A, B). Shown are the percentages of animals with over-retracted, partially retracted, or unretracted tails. n ≥ 100, except for *lin-41(ma104)* (n = 80). Worms were grown at 25°C. **(B, C)** *lin-41(ma104)* animals display a precocious male tail retraction phenotype and were included as a control.

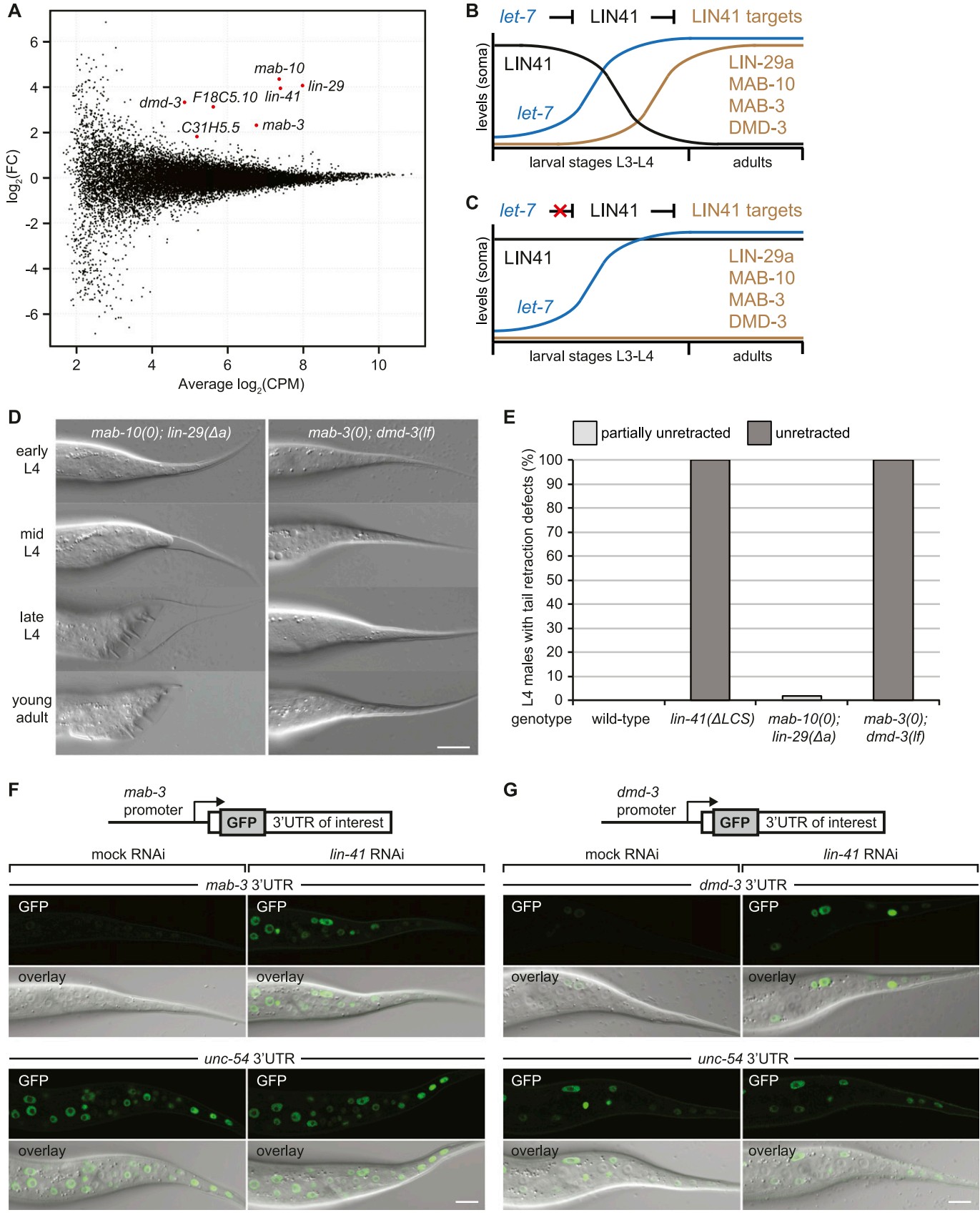

**A**

**B** *let-7* ⊣ LIN41 ⊣ LIN41 targets

**C** *let-7* ✕⊣ LIN41 ⊣ LIN41 targets

**D** *mab-10(0); lin-29(Δa)* | *mab-3(0); dmd-3(lf)*

early L4 | mid L4 | late L4 | young adult

**E** partially unretracted | unretracted

genotype: wild-type | *lin-41(ΔLCS)* | *mab-10(0); lin-29(Δa)* | *mab-3(0); dmd-3(lf)*

**F** *mab-3* promoter — GFP — 3'UTR of interest

mock RNAi | *lin-41* RNAi

*mab-3* 3'UTR

GFP | GFP
overlay | overlay

*unc-54* 3'UTR

GFP | GFP
overlay | overlay

**G** *dmd-3* promoter — GFP — 3'UTR of interest

mock RNAi | *lin-41* RNAi

*dmd-3* 3'UTR

GFP | GFP
overlay | overlay

*unc-54* 3'UTR

GFP | GFP
overlay | overlay

We performed three independent experiments and determined a set of LIN41-bound mRNAs using edgeR (Robinson et al, 2010) with a false discovery rate of <0.05. We identified seven mRNAs as reproducibly associated with LIN41: *lin-29a, mab-10, mab-3, dmd-3, lin-41*, and the unnamed transcripts *F18C5.10* and *C31H5.5* (Fig 3A). Gratifyingly, this list included all four previously identified mRNA targets of LIN41, namely *lin-29a, mab-10, mab-3*, and *dmd-3* (Aeschimann et al, 2017), and surprisingly few additional transcripts. Binding of LIN41 to its own mRNA may suggest an autoregulatory function, as previously proposed to occur on the post-translational level (Del Rio-Albrechtsen et al, 2006). Alternatively, it may stem from the immunoprecipitation of nascent FLAG::GFP::LIN41 protein, still bound to the translating ribosome and thus its own mRNA. For the remaining two putative targets, *F18C5.10* and *C31H5.5*, we find no evidence in the available data (Aeschimann et al, 2017) that LIN41 affects their translation rates or mRNA stability. Therefore, we hypothesized that LIN41 would execute its physiological functions through the repression of only four target mRNAs; that is, *lin-29, mab-10, mab-3*, and *dmd-3*.

Previously, we showed that all four targets accumulate shortly before wild-type worms turn into adults, when *let-7* has silenced LIN41 sufficiently to prevent repression of LIN41 targets ([Aeschimann et al, 2017] and schematically depicted in Fig 3B). Conversely, in animals in which *let-7*–mediated silencing of LIN41 is lost, sustained LIN41 accumulation keeps the four LIN41 targets repressed (Fig 3C). Hence, if our hypothesis were true, reducing the activity of these four LIN41 targets should recapitulate the phenotypes of *let-7(PM)* and *lin-41(ΔLCS)*–mutant animals.

We noticed that we could group the four LIN41 targets into two pairs of transcriptional regulators; that is, LIN-29 + MAB-10 and MAB-3 + DMD-3, respectively. This is because MAB-10, orthologous to mammalian NAB1/2, is a cofactor of LIN-29, which is an early growth response (EGR)-type transcription factor of the *Krüppel* family (Harris & Horvitz, 2011). MAB-3 and DMD-3, on the other hand, are both DM (Doublesex/MAB-3) domain–containing transcription factors, proposed to act at least partially redundantly on common targets due to similar binding motifs (Yi & Zarkower, 1999; Mason et al, 2008).

To explore the effects of losing the activity of LIN41 targets, we created deletion alleles of *mab-10* and *mab-3* by removing almost the entire coding region using clustered regularly interspaced short palindromic repeats (CRISPR)-Cas9 (Fig S1A). *lin-29* encodes two major protein isoforms, LIN-29a and LIN-29b, which are thought to function redundantly and both of which can bind MAB-10 (Bettinger et al, 1997; Harris & Horvitz, 2011; Rougvie & Ambros, 1995). Since we

previously established that LIN41 targets only the *lin-29a* isoform but not *lin-29b* (Aeschimann et al, 2017), we created an allele that specifically disrupts *lin-29a* (Fig S1A). For *dmd-3*, we used the previously published *dmd-3(ok1327)* allele, which is considered a null allele for its function in male tail development (Mason et al, 2008), and to which we refer here as *dmd-3(lf)*.

## LIN41 regulates *mab-3* and *dmd-3* to time male tail retraction

First, we examined male tail retraction. Male mating deficiencies and abnormalities in the tail tips have been reported for each of the four LIN41 targets as single mutants (Hodgkin, 1983; Euling et al, 1999; Mason et al, 2008). However, whereas cell retraction appears to occur normally in *lin-29* and *mab-10* single mutant males (Hodgkin, 1983; Euling et al, 1999), *mab-3* and *dmd-3* single mutant males exhibit Lep phenotypes. These are weak and of low penetrance for *mab-3* but stronger and highly penetrant for *dmd-3* (Mason et al, 2008). Moreover, *mab-3; dmd-3* double mutant males have completely unretracted male tails (Mason et al, 2008). To compare phenotypes of *lin-41(ΔLCS)* mutants with those of the LIN41 target pairs, we observed and quantified the male tail phenotypes of *mab-10(0) lin-29(Δa)* as well as of *mab-3(0); dmd-3(lf)* double mutant animals. *mab-10(0) lin-29(Δa)* double mutant males exhibited normal tail cell retraction, although they displayed shorter fan and ray structures than wild-type animals (Figs 3D and E, and S1B and Table S3). By contrast, *mab-3(0); dmd-3(lf)* males were completely deficient for cell retraction at any stage during development (Figs 3D and E, and S1B and Table S3). Hence, combined mutation of *mab-3* and *dmd-3* is phenotypically equivalent to the *lin-41(ΔLCS)* and the *let-7(PM)* single mutations (Figs 1F, 2A and C and Table S3), suggesting that *let-7* promotes the morphogenesis of male tails predominantly by activating *mab-3* and *dmd-3*.

## LIN41 regulates *mab-3* and *dmd-3* directly and post-transcriptionally to control tail morphogenesis

Previously, it was proposed that *dmd-3* expression was indirectly regulated by LIN41, through an unknown mechanism involving the *dmd-3* promoter (Mason et al, 2008). However, by reporter gene assays in the hermaphrodite epidermis, we showed that both *mab-3* and *dmd-3* 3′ UTRs can confer LIN41-dependent regulation (Aeschimann et al, 2017). To test whether physiologic regulation of *mab-3* and *dmd-3* by LIN41 in the male tail epidermis was promoter-dependent or 3′ UTR-dependent, we created reporter lines to

---

**Figure 3. LIN41 controls male tail cell retraction through MAB-3 and DMD-3.**
**(A)** MA plot for anti-FLAG RIP-seq experiments with FLAG::GFP::LIN41 and FLAG::GFP::SART-3 as a control. Semi-synchronous L3/L4 stage worm populations enriched in males (*him-5(e1490)* genetic background) were compared in three independent biological replicates. The plot compares the fold change (FC) in IP-to-input enrichments for RNA-sequencing reads in the FLAG::GFP::LIN41 versus the FLAG::GFP::SART-3 IP (y-axis) with the mean mRNA abundance (x-axis, CPM, counts per million). Genes passing the cutoff of FDR < 0.05 are highlighted in red and labeled. **(B, C)** Schematic depiction of the expression patterns of the LIN41 protein, the *let-7* miRNA, and the LIN41 target proteins in the soma during development from larvae to adult worms, in the wild-type situation (B) and when *let-7* fails to repress *lin-41* (C). **(D)** Example micrographs of male tails at different developmental time points in the indicated genetic backgrounds. Scale bar: 20 μm. **(E)** Quantification of the male tail phenotypes at the late L4 larval stage of *mab-10(0) lin-29(Δa)* and *mab-3(0); dmd-3(lf)* animals. The data for wild-type and *lin-41(ΔLCS)* males are re-plotted from Fig 1 for reference. Shown are the percentages of animals with over-retracted, partially retracted or unretracted tails. n ≥ 100, worms were grown at 25°C.
**(F, G)** Confocal images of the male tail epidermis in young L3-stage male animals expressing nuclear-localized GFP(PEST)::H2B reporters, driven from the *mab-3* (F) and *dmd-3* (G) promoters and fused to their orthologous 3′ UTR sequences or the unregulated *unc-54* 3′ UTR as indicated. Animals were grown for 20 h at 25°C on *lin-41* or mock RNAi bacteria. Scale bars: 10 μm.

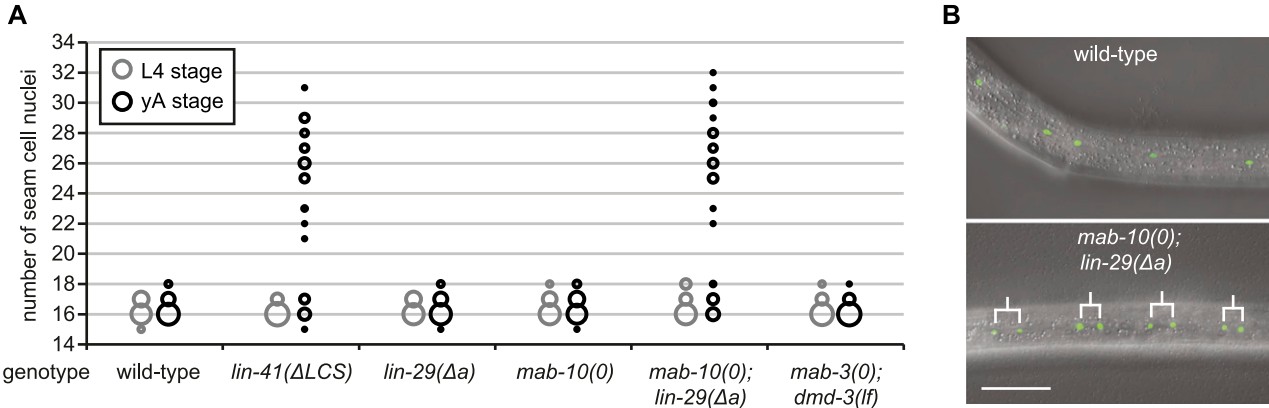

**Figure 4. LIN41 controls self-renewal of seam cells through LIN-29a and its co-factor MAB-10.**
**(A)** Quantification of seam cell nuclei of L4 stage and young adult (yA) animals of the indicated genetic backgrounds, as in Fig 1E. n = 20 for L4, n > 50 for yA, worms grown at 25°C. Results for wild-type and *lin-41(ΔLCS)* animals are re-plotted from Fig 1 for reference. **(B)** Example micrographs of a young adult wild-type worm and a worm lacking LIN-29a and MAB-10 expressing transgenic *scm::gfp*. Branched lines indicate seam cell nuclei originating from extra cell divisions. Scale bar: 50 μm.

express *gfp(pest)::h2b* from putative *mab-3* and *dmd-3* promoters, that is, the 4-kb region upstream of their start codons. For each promoter, we created two reporters, one with a promoter-orthologous 3′ UTR that harbors LIN41 target sites and one with the heterologous *unc-54* 3′ UTR that is not regulated by LIN41 (Aeschimann et al, 2017).

All four reporters were expressed in the tail epidermis and other tissues of males at the L4 stage. However, differences occurred in early L3-stage males, when LIN41 is present: GFP accumulation in most tissues, and in particular in the epidermis, was almost undetectable for both reporters carrying the promoter-orthologous 3′ UTRs (Fig 3F and G). By contrast, both reporters containing the heterologous *unc-54* 3′ UTR yielded strong GFP accumulation in various cells, including the epidermal cells of the tail region (Fig 3F and G and data not shown). Moreover, for both reporters with the promoter-orthologous 3′ UTRs, depletion of LIN41 by RNAi resulted in strong GFP signals in many tissues, including the epidermal cells of the tail region (Fig 3F and G). We conclude that temporal control of MAB-3 and DMD-3 accumulation is predominantly conferred by post-transcriptional regulation, through LIN41-binding to their 3′ UTRs and, hence, that LIN41 regulates the timing of cell retraction in the male tail through *mab-3* and *dmd-3* mRNAs as direct targets.

### Seam cell exit from the cell cycle is controlled by LIN41-mediated regulation of *lin-29a* and *mab-10*

Next, we asked how the repression of *lin-41* triggers the cell cycle exit of seam cells upon transition to adulthood. Because *lin-29* and *mab-10* have both been implicated in seam cell development (Ambros & Horvitz, 1984; Harris & Horvitz, 2011), we wondered if we could phenocopy *let-7(PM)* or *lin-41(ΔLCS)* mutants by mutating *lin-29a* and/or *mab-10*. Seam cell numbers in young adults of either *lin-29(Δa)* or *mab-10(0)* single mutants were unchanged from the wild-type situation (Fig 4A and Table S2), although older *mab-10(0)* adults displayed a few additional seam cells (data not shown and Harris & Horvitz, 2011). By contrast, young adult *mab-10(0) lin-29(Δa)* double mutant animals displayed additional seam cell divisions to a comparable extent as

*let-7(PM)*– or *lin-41(ΔLCS)*–mutant animals (Figs 1D and 4 and Table S2). Depletion of both MAB-3 and DMD-3 did not affect seam cell numbers (Fig 4A and Table S2). We conclude that *lin-29a* and *mab-10* are a major, likely sole, output of the *let-7*–LIN41 regulatory module for control of seam cell self-renewal.

We propose that the synergistic seam cell phenotype of *mab-10(0)* and *lin-29(Δa)* mutations is most parsimoniously explained by MAB-10 functioning with both LIN-29a and LIN-29b to control seam cell self-renewal. Indeed, *lin-29(xe37)*–mutant animals, which lack both LIN-29a and LIN-29b (Fig S1) and which we therefore designated *lin-29(0)*, display highly penetrant extra seam cell divisions at the young adult stage (Table S2), consistent with an earlier report for putative *lin-29(0)*–mutant animals (Ambros & Horvitz, 1984). Hence, it appears that LIN41 controls seam cell fates by regulating LIN-29a activity directly, by decreasing its protein levels, and by modulating LIN-29b activity indirectly, and presumably less extensively, by decreasing the levels of its cofactor MAB-10.

### Inactivation of the four LIN41 targets recapitulates *let-7*–mutant vulval bursting only partially

Our findings thus far show that continued silencing of two distinct target pairs of LIN41 explains *let-7*–mutant phenotypes in male tail morphogenesis and seam cell self-renewal. Therefore, we were surprised to find that inactivation of LIN41 targets did not easily explain the most obvious *let-7*–mutant phenotype, highly penetrant vulval bursting and death (Fig 5A and Table S1): Even among *mab-3(0); mab-10(0) lin-29(Δa); dmd-3(lf)* quadruple mutant animals, only ~15% burst, a bursting frequency greatly below the ~90% penetrance seen with *let-7(PM)* and *lin-41(ΔLCS)* single mutations. Examination of LIN41 target single mutations as well as of the relevant mutation pairs, that is, *mab-3(0); dmd-3(lf)* and *mab-10 lin-29(Δa)*, showed that the bursting frequency of 15% was caused by the combined lack of LIN-29a and MAB-10.

These results might suggest that *let-7*–LIN41 promote vulval integrity through additional, currently unknown LIN41 targets or functions. However, we favored a different explanation, namely,

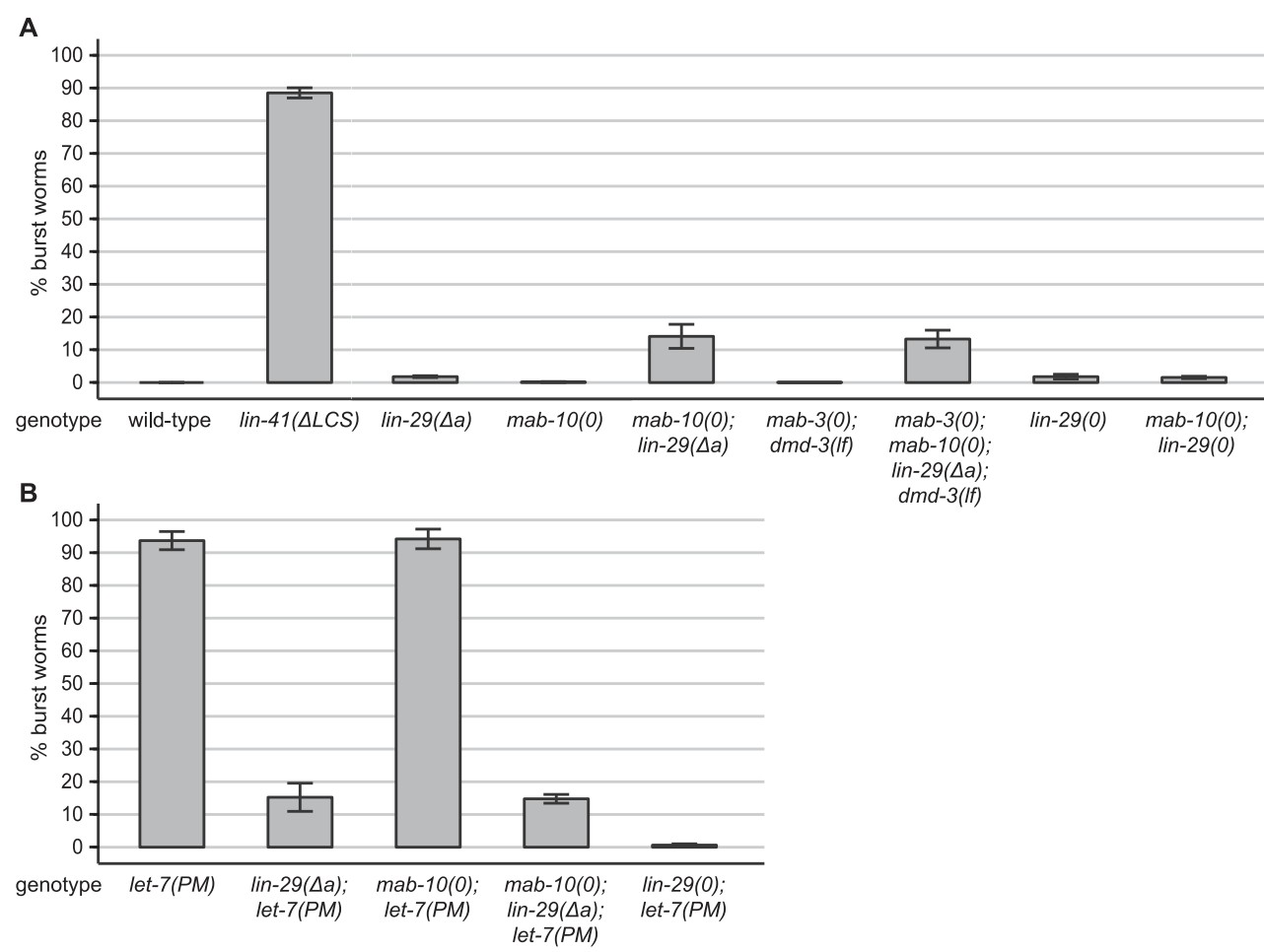

**Figure 5. LIN-29a and MAB-10 are involved in the vulva bursting phenotype.**
**(A, B)** Quantification of burst worms of the indicated genotypes, grown in a synchronized population at 25°C for 45 h. Data as mean ± SEM of N = 3 independent biological replicates with n ≥ 400 worms counted for each genotype and replicate. In panel (A), results for wild-type and *lin-41(ΔLCS)* animals are re-plotted from Fig 1 for reference.

that highly penetrant vulval bursting required residual or tissue-specific activity of LIN-29 and/or its cofactor MAB-10, whereas a complete loss of their activity caused by gene deletions leads to a much lower penetrance. This is because *mab-10(0) lin-29(Δa); let-7 (PM)* triple mutant animals phenocopied the *mab-10(0) lin-29(Δa)* double mutant rather than the *let-7(PM)* single mutant phenotype (Fig 5B and Table S1). In other words, mutations in *mab-10* and *lin-29a* suppress the *let-7(PM)*–mutant phenotype. Moreover, loss of both *lin-29a* and *lin-29b* in *lin-29(0)*–mutant animals caused bursting in only < 2% of animals, and it was not further enhanced by additional loss of *mab-10* (Fig 5A and Table S1). In fact, loss of both LIN-29 isoforms reduced bursting due to the *let-7(PM)* mutation to < 2% in *lin-29(0); let-7(PM)* double mutant animals (Fig 5B and Table S1). These findings suggest that loss of all or most LIN-29 activity is incompatible with penetrant vulval rupturing.

### Expression of *lin-29a* in the AC is not regulated by *let-7* and affects the integrity of the vulva-uterine tract

Further investigation revealed that the *lin-29(Δa)* mutation alone reduced the bursting frequency of *let-7(PM)* animals to about 15%,

whereas the *mab-10(0)* mutation did not reduce the bursting frequency (Fig 5B and Table S1). Hence, it appeared that a residual or tissue-specific LIN-29a activity is necessary to permit vulval bursting with high penetrance. To determine when and where *lin-29a* is differentially expressed in wild-type versus *let-7(PM)* worms, we specifically tagged the endogenous LIN-29a isoform by placing a GFP::3xFLAG tag at the N terminus (Fig S1A). As expected, we observed *lin-29a* expression in the epidermis of wild-type but not *let-7 (PM)*–mutant late L4-stage animals (Fig 6A). However, not all *lin-29a* expression was lost in the *let-7(PM)* mutant. Specifically, we observed GFP::LIN-29a accumulation in the distal tip cell, the anchor cell (AC), and most vulval cells. The strongest GFP signal was that in the AC (Fig 6B), observed from about the L2-to-L3 molt onwards.

The AC establishes the uterine-vulval connection through which *let-7*–mutant animals burst. Moreover, and as discussed in more detail in the following section, *lin-29* mutations cause defects in vulval-uterine cell fates that are AC-dependent (Newman et al, 2000) and that are not shared by *let-7*–mutant animals (Ecsedi et al, 2015). Therefore, we wondered if LIN-29a accumulation in the AC of *let-7(PM)* but not *mab-10(0) lin-29(Δa)* double mutant and *lin-29(0)* single mutant animals could explain why the former, but not the

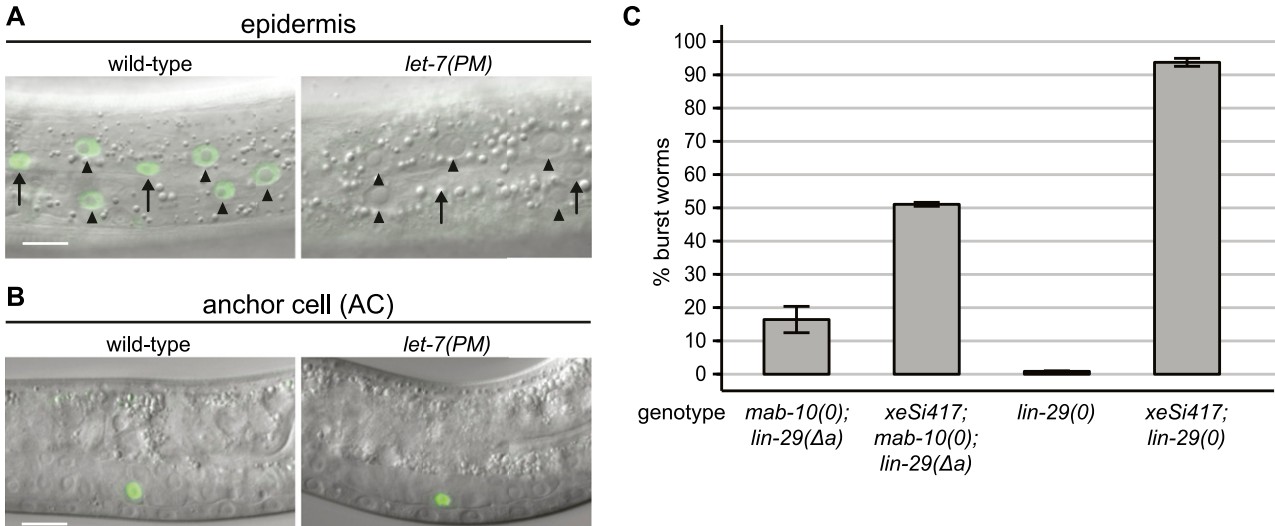

**Figure 6. LIN-29a accumulation in the AC is independent of *let-7* and increases the penetrance of vulval bursting.**
**(A, B)** Example micrographs of worms expressing GFP-tagged LIN-29a, showing the epidermis of late L4-stage animals (A) and the anchor cell of L3-stage animals (B). Scale bar: 10 μm. Arrows point to seam cell nuclei, arrowheads to hyp7 nuclei. **(C)** Quantification of burst worms of the indicated genotypes, grown in a synchronized population at 25°C for 45 h. The *xeSi417* transgene drives LIN-29a accumulation specifically in the anchor cell. Data as mean ± SEM of N = 3 independent biological replicates with n ≥ 400 worms counted for each genotype and replicate.

two latter, exhibited a highly penetrant bursting phenotype. To investigate this, we produced LIN-29a specifically in the AC of *mab-10(0) lin-29(Δa)*– and *lin-29(0)*–mutant animals. We used a single-copy integrated *xeSi417* transgene that combines the Δ*pes-10* basal promoter (Harfe & Fire, 1998) with an AC-specific enhancer of *lin-3* (ACEL, [Hwang & Sternberg, 2004]) to express an operon containing *lin-29a* followed by nuclear-localized *gfp* to assess the expression pattern. We observed consistent and specific AC expression (Fig S2), although the GFP levels failed to reach those of the endogenously tagged GFP::LIN-29a protein.

Wild-type animals expressing the transgene did not reveal any overt defects. However, consistent with our hypothesis, expression of this transgene increased the frequency of bursting to about 50% in *mab-10(0) lin-29(Δa)* double mutant animals, and to >90% in *lin-29(0)*–mutant animals (Fig 6C). We conclude that animals producing LIN-29a in the AC but otherwise lacking LIN-29a and its cofactor MAB-10, or LIN-29 activity altogether, are prone to vulval rupturing. Hence, this finding suggests that *let-7(PM)*–mutant animals die because of a lack of LIN-29 activity in some tissues while retaining LIN-29a activity in the AC.

### *lin-29a* expression in the AC promotes uterine seam cell formation

Given the above results, we sought to examine the role of LIN-29a in the AC. During wild-type development, the AC induces and coordinates cell fates of both the vulva and the uterus. In uterine development, the AC specifies its lateral neighbors as π cells, and some of the π daughter cells fuse with each other and eventually with the AC to form the uterine seam cell (utse). The utse is important for the structure of the egg-laying apparatus, as it anchors the adult uterus to the epidermal seam. It also forms a thin cytoplasmic extension that separates the lumens of vulva

and uterus until it is broken when the first embryo is laid. In *lin-29(0)* mutants, π cell fates are not specified and the utse does not form (Newman et al, 2000). Instead, the AC remains unfused, and the vulva and uterus are separated by thick tissue rather than the thin utse (Newman et al, 2000) (Fig 7A). This phenotype is not shared by *let-7*–mutant animals (Ecsedi et al, 2015). Instead, when *let-7*–mutant animals die by bursting through the vulva, the intestine is pushed out of the animal, which breaks the thin utse. Hence, we hypothesized that it was the presence of the thick tissue separating the vulva and the uterus that prevented vulval rupturing in *lin-29(0)* single mutant and in *mab-10(0) lin-29(Δa)* double mutant animals. We further hypothesized that re-expression of *lin-29a* in the AC would restore utse formation, and thereby render animals that lack LIN-29 activity in other tissues prone to bursting.

In support of these hypotheses, we found a thick tissue separating the vulval and uterine lumens in >95% of *lin-29(0)* single mutant hermaphrodites and in >50% of *mab-10(0) lin-29(Δa)* double mutant hermaphrodites. Consistent with an impairment of vulval function, these animals displayed highly penetrant egg-laying deficient (Egl) and protruding vulvae (Pvl) phenotypes (Fig S3). Production of LIN-29 in the AC, using the *xeSi417* transgene, restored utse formation in all animals of either mutant strain as scored by re-appearance of the thin cytoplasmic extension (Fig 7A and B and Table S4).

We conclude that expression of *lin-29a* in the AC contributes to wild-type utse formation, and that in the presence of a wild-type utse, lack of LIN-29 activity in tissues other than the AC causes animal bursting. In other words, *mab-10(0) lin-29(Δa)* double mutant and *lin-29(0)* single mutant animals, like *let-7*–mutant animals, are prone to rupturing, but observation of this phenotype is obscured when lack of *lin-29a* expression in the AC prevents normal utse formation. Hence, the *let-7* bursting phenotype can be

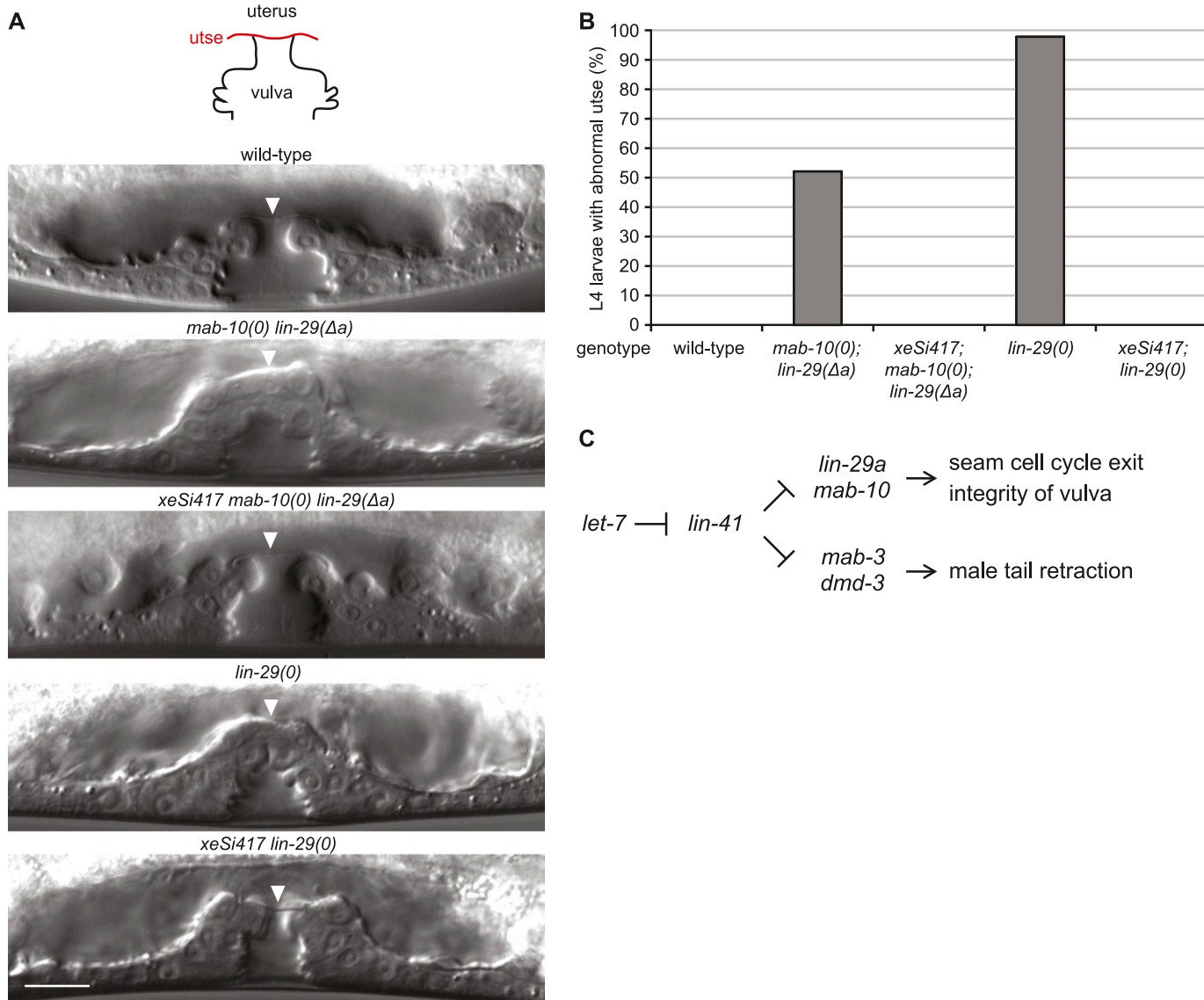

**Figure 7. LIN-29a accumulation in the AC is required for utse formation.**
**(A)** Representative micrographs of the late L4-stage vulva in wild-type animals and *mab-10(0) lin-29(Δa)*– or *lin-29(0)*–mutant animals with and without AC production of LIN-29a from the *xeSi417* transgene. Arrowheads point to the thin or thick tissue separating the vulva from the uterus. Scale bar: 10 *μm*. **(B)** Quantification of uterine-vulva connection phenotypes of late L4 larvae of the indicated genotypes. n ≥ 40, worms grown at 25°C. **(C)** Model for the output of the *let-7*–LIN41 pathway regulating two pairs of LIN41 targets involved in transcription. LIN-29a and MAB-10 stop seam cell divisions and prevent vulval rupturing. MAB-3 and DMD-3 drive cell retraction events to shape the male tail.

explained by sustained LIN41-mediated repression of *mab-10* and *lin-29a* in tissues other than the AC.

## Discussion

The *C. elegans* heterochronic pathway is arguably one of the best-characterized developmental timing pathways, and recent evidence has suggested that its function in controlling the onset of J/A transition might be evolutionarily conserved (Corre et al, 2016; Faunes & Larrain, 2016; Pereira et al, 2019). However, despite extensive knowledge of the genetic players and their relative

positions in the heterochronic pathway, a detailed mechanistic concept is still lacking and requires better characterization of direct molecular connections among individual heterochronic genes. To begin filling this gap, we have focused here on elucidating the molecular mechanisms that time transition into adulthood by identifying and functionally characterizing the players immediately downstream of *let-7*.

Extending our previous finding that *let-7* prevents vulval rupturing through regulation of only a single target (Ecsedi et al, 2015), LIN41, we find that the function of *let-7* in additional tissues and processes also relies on regulation of only LIN41. This finding is consistent with a previous gene expression analysis, which

revealed that global, animal-wide changes in gene expression through *let-7* inactivation are recapitulated well by impairing selectively its silencing of only *lin-41* (Aeschimann et al, 2017). Yet, it contrasts with previous reports that identified numerous putative *let-7* targets (Abrahante et al, 2003; Lin et al, 2003; Großhans et al, 2005; Andachi, 2008; Jovanovic et al, 2010; Hunter et al, 2013). However, those reports relied on circumstantial evidence to establish miRNA targets, typically suppression of certain *let-7*–mutant phenotypes through mutation or depletion of suspected target genes. The advent of genome editing has now enabled a direct analysis of the physiological relevance of an individual target, by specifically uncoupling it from *let-7*, or recoupling it to a mutant variant of *let-7*. The finding that the three *let-7*–regulated processes that we investigated—vulval rupturing, male tail morphogenesis, and seam cell fate control—are all dependent on LIN41 as the key *let-7* target, now establishes *let-7*–LIN41 as a versatile regulatory module. Indeed, *let-7* was recently found to control the timing of sexually dimorphic neuron differentiation in male *C. elegans*, and this function as well appears to rely solely on its ability to regulate LIN41 (Pereira et al, 2019).

Although miRNAs are often thought to act through a network activity where they silence many targets modestly but coordinately (Bracken et al, 2016), other instances have been reported, both in *C. elegans* and mice, where only one or two targets appear to explain physiological functions of miRNAs (Lee et al, 1993; Wightman et al, 1993; Moss et al, 1997; Johnston & Hobert, 2003; Dorsett et al, 2008; Teng et al, 2008; Lu et al, 2015; Drexel et al, 2016). Hence, it remains to be determined whether the network activity model accurately describes the predominant mode of miRNA function in animals.

Downstream of LIN41 in the heterochronic pathway, our results define the immediate next layer of regulatory function by identifying the relevant direct targets of LIN41 and their specific developmental roles. The results presented here and elsewhere (Pereira et al, 2019) demonstrate that *lin-29a*, *mab-10*, *mab-3*, and *dmd-3*, previously shown to be directly, post-transcriptionally silenced by LIN41 (Aeschimann et al, 2017), act as the main regulatory output of the *let-7*–LIN41 pathway in the J/A transition. Thus, despite other proposed molecular activities as an E3 ubiquitin ligase and a structural protein, the function of LIN41 as an RNA-binding protein accounts for its known heterochronic functions. The number of relevant LIN41 mRNA targets is unexpectedly small, particularly given previous reports that showed rather promiscuous RNA-binding activity of LIN41 in the adult hermaphrodite germline (Tsukamoto et al, 2017; Kumari et al, 2018). Whether more selective mRNA binding by LIN41 in the larval soma is a consequence of the differences in LIN41 protein concentration in larval soma versus adult germline, or of a function in different protein complexes in each situation remains an open question.

Whereas *let-7*–LIN41 act sequentially, in a linear fashion, to control transition to adulthood, the heterochronic pathway branches at the point of LIN41 output (Fig 7C). This architecture facilitates a coordinated and timely activation of different developmental events as LIN41 becomes silenced through increasing *let-7* levels in late stage larvae. The LIN41 targets appear to be grouped into two functionally separated parallel pathways, each with a distinct pair of transcriptional regulators: DMD-3 + MAB-3 mediate male tail retraction, and LIN-29a + MAB-10 promote both vulval integrity and cessation of seam cell self-renewal.

Within each pair, the individual factors have partially redundant functions, as individual mutations cause either no, or only partially penetrant phenotypes. By co-regulating partially redundant genes within each pathway, LIN41 itself assumes a more unique function and, as a consequence, elevated LIN41 levels as in *lin-41(ΔLCS)* mutants lead to fully penetrant phenotypes. Thus, to control events for the transition to adulthood, a cascade of post-transcriptional regulators eventually times the expression of two pairs of transcriptional regulators.

The essential and overlapping function of *dmd-3* and *mab-3* in male tail cell retraction had been reported previously (Mason et al, 2008). However, it was unclear to what extent this would explain *let-7* function because previously reported *let-7*–mutant phenotypes were weaker than those of *mab-3; dmd-3* double mutant animals (Del Rio-Albrechtsen et al, 2006; Mason et al, 2008). Moreover, LIN41 had been shown to regulate *dmd-3* expression, but regulation was thought to occur transcriptionally and indirectly, through the function of an unknown direct target of LIN41 (Mason et al, 2008). We can now reconcile these results by showing that complete inactivation of *let-7*–mediated silencing of LIN41 recapitulates the severe *mab-3; dmd-3* double mutant phenotypes and that LIN41 regulates both *dmd-3* and *mab-3* directly and post-transcriptionally.

Among the three mutant phenotypes that we have investigated, vulval rupturing was the most enigmatic. Although the phenotype was first described nearly two decades ago (Reinhart et al, 2000), its cause has remained unclear. As we now show, combined dysregulation of *lin-29a* and *mab-10* is an important factor. Specifically, sustained *lin-29a* expression in the AC appears to combine with the absence of *lin-29a* and *mab-10* in other, yet to be determined cells of the vulval-uterine system and/or the epidermis to cause penetrant vulval bursting. The accumulation of the LIN-29a protein in the AC is sufficient to promote utse formation, at least at the morphological level, which may be needed for vulval bursting to occur. As 50% of *mab-10(0) lin-29(Δa)* animals that produced LIN-29a in the AC burst, but 100% exhibited overtly normal utse morphology, a thin utse alone appears insufficient to permit bursting, and it remains an open question what prevents a fully penetrant phenotype in these animals. However, the thick cell layer that separates uterus and vulva in *lin-29(0)*– or *mab-10(0) lin-29(Δa)*–mutant animals may be sufficient to prevent bursting, as it is presumably more resistant than a wild-type utse to the internal pressure in the worm. Taken together, our findings reveal an important role of MAB-10 and LIN-29a as effectors of *let-7*–LIN41 in its function to ensure vulval integrity, and they explain why *lin-29* and *let-7* mutations were previously found to yield incompatible phenotypes (Ecsedi et al, 2015).

What are then the events that fail and thereby cause vulval bursting when *lin-29a* and *mab-10* are not properly up-regulated in *let-7*–mutant animals? No obvious defects in vulval or uterine development of *let-7(PM)* animals could be detected (Ecsedi et al, 2015). This could suggest that a *let-7* function in a tissue other than the vulva or uterus is crucial for vulval integrity. However, previous work also revealed that in *let-7(PM)* animals, re-expression of *let-7*

in the epidermis, uterus, and vulva sufficed to prevent vulval bursting, whereas re-expression in the epidermal hyp7 syncytium only was sufficient to restore some degree of epidermal differentiation but was insufficient to prevent vulval bursting (Ecsedi et al, 2015). Hence, although a lack of suitably specific expression tools prevented a more refined dissection of the spatial requirements of *let-7* expression, it is likely that expression of *let-7* in the vulva, uterus, and/or epidermal seam is required for vulval integrity. Therefore, it seems possible that attachments of uterine and/or vulval cells to each other and/or to the lateral seam are defective in *let-7*–mutant animals. We expect that a detailed analysis of the expression patterns of *lin-29a* and *mab-10*, and the specific tissues and cell-types that require *let-7* expression for vulval integrity may help define where *let-7*–LIN41 regulate *lin-29a* and *mab-10* expression to prevent vulval rupturing. Future studies aimed at uncovering the transcriptionally regulated target genes of LIN-29a+MAB-10 may additionally provide insight into the underlying defects, and this and the study of MAB-3+DMD-3 targets will illuminate how *let-7*–LIN41 direct cell fate and morphological changes at the transition to adulthood.

Our study focused on the functions of the *let-7*–LIN41 pathway in controlling self-renewal and transition to adulthood in *C. elegans*, yet these functions may be phylogenetically conserved. Specifically, the mammalian homologs of *let-7*, LIN41, and the *let-7* regulator LIN28 are known to be involved in the control of self-renewal versus differentiation programs in different cell types such as embryonic stem cells or neural progenitor cells (Faunes & Larrain, 2016). EGR and NAB proteins, homologous to the LIN41 targets MAB-10 and LIN-29a that we have shown here to be essential for these programs in *C. elegans* seam cells, have not been mechanistically linked to this pathway. However, they are crucial regulators of proliferation and/or terminal differentiation programs in different mammalian cell types, including embryonic stem cells, different types of blood cells, or Schwann cells (Nguyen et al, 1993; Topilko et al, 1994; Le et al, 2005; Laslo et al, 2006; Min et al, 2008; Du et al, 2014; Worringer et al, 2014). Moreover, EGR1 overexpression antagonizes somatic cell reprogramming promoted through LIN41 (Worringer et al, 2014), although we note that EGR1 does not appear to be the closest homologue of LIN-29a (Pereira et al, 2019).

Yet more intriguingly, timing defects in human puberty have been linked to genetic variations in LIN28 (Faunes & Larrain, 2016), and in mice, both *Lin28(gf)* and *let-7(lf)* mutations were reported to delay the onset of puberty (Zhu et al, 2010; Corre et al, 2016). Whether mammalian LIN41 is involved in regulating the timing of puberty remains to be tested, and gene expression changes that control puberty remain poorly understood. However, we note that homologs of the transcriptional regulators downstream of LIN41 have known roles in the development of sex-specific structures in other animals including mammals. For example, mice homozygous for a null mutant of EGR1 are sterile, with females having an abnormal development of the ovary (Topilko et al, 1998). Moreover, DM domain–containing transcription factors control sexual differentiation across evolution and have crucial roles in the development of mammalian testis and germline (Kopp, 2012; Zhang & Zarkower, 2017). Given these tantalizing hints and apparent similarities, we suggest that it will be relevant to study whether LIN28 and *let-7* control the timing of mammalian sexual organ development, and

possibly other puberty-related events, through LIN41 and EGR/NAB or DM domain–containing transcription factors.

# Materials and Methods

## *C. elegans*

Worm strains used in this study are listed in Table S5. Bristol N2 was used as the wild-type strain. Animals were synchronized as described (Aeschimann et al, 2017) and grown on 2% NGM agar plates with *Escherichia coli* OP50 bacteria (Stiernagle, 2006) unless specified otherwise. For RIP-Seq experiments, worms were grown on enriched peptone plates with *E. coli* NA22 bacteria (Evans, 2006). For RNAi experiments, arrested L1s were plated on RNAi-inducing NGM agar plates with *E. coli* HT115 bacteria containing plasmids targeting the gene of interest (Ahringer, 2006).

### Generation of novel *lin-29*, *lin-29a*, *mab-10*, and *mab-3* null mutant alleles using CRISPR-Cas9

Wild-type worms were injected with a mix containing 50 ng/µl of pIK155, 100 ng/µl of pIK198 with a cloned template for sgRNA1, 100 ng/µl of pIK198 with a cloned template for sgRNA2, 5 ng/µl pCFJ90, and 5 ng/µl pCFJ104, as previously described (Katic et al, 2015). Single F1 progeny of injected wild-type worms were picked to individual plates and the F2 progeny screened for deletions using PCR assays. After analysis by DNA sequencing, the alleles were outcrossed three times to the wild-type strain.

To obtain mutant alleles for *lin-29*, *mab-10*, and *mab-3* that are undoubtedly null for the encoded protein, two single guide RNAs (sgRNAs) per gene were injected to generate large deletions spanning almost the entire coding region. The following pairs were used: (i) *lin-29* sgRNA1: gctggaaccaccactggctc, *lin-29* sgRNA2: atat-tatttatcagtgattg; (ii) *mab-10* sgRNA1: gatgatgatgatgaagaggt, *mab-10* sgRNA2: gctcccggaatcttgaagct; and (iii) *mab-3* sgRNA1: aggagctc-taatgctcaccg, *mab-3* sgRNA2: agctcagctcaatttgggcg.

For *lin-29*, we generated a large deletion spanning exons 2–11 and thus most of the coding region of both *lin-29a* and *lin-29b*. Specifically, the resulting *lin-29(xe37) = lin-29(0)* allele is a 14,801-bp deletion with a 2-bp insertion with the following flanking sequences: 5′ ggactctggaatagctggaa—*xe37* deletion—*xe37* insertion (aa)—aatatgaaaaatcattccta 3′. Translation of *xe37* yields only a short stretch of 28 amino acids (MDQTVLDSAFNSPVDSGIAG-KNMKNHSY*), containing the N-terminal 20 and the C-terminal 7 amino acids of LIN-29a. The small insertion leads to translation of an additional lysine (K).

For *mab-10*, the resulting *mab-10(xe44) = mab-10(0)* allele spans exons 3–9 and is a 2,901-bp deletion with a 4-bp insertion with the following flanking sequences: 5′ ttatcatctcttacaactca—*xe44* deletion—*xe44* insertion (ctct)—tatttttttgttttcctc<u>gtga</u> 3′. Translation of *xe44* yields a 58-amino acid stretch (MSSSSSSSLPTSSASTTTSSITSRPSASHHLESILSSSSSS-PSILSSLTT-HSYFLFSS*) containing the N-terminal 50 amino acids of MAB-10, followed by eight additional amino acids, translated from the small insertion and the *mab-10* 3′ UTR, and a stop codon (underlined in the flanking sequence above).

For *mab-3*, the resulting *mab-3(xe49) = mab-3(0)* allele is a 4,291-bp deletion starting in exon 2, just downstream of the ATG start codon of the longer isoform, and ending downstream of the stop codon. The flanking sequences are: 5′ tttgcagaggagctctaatg—*xe49* deletion—ctccgcccacactttcccag 3′. Translation of *xe49* is presumably initiated at the normal ATG start codon, but then translates a sequence that is normally noncoding, yielding a stretch of 31 amino acids (MLRPHFPRITVLFLALRLSFSFPLSLFYLGK*) unrelated to the MAB-3 protein.

To specifically mutate *lin-29a* without affecting expression of *lin-29b*, we deleted part of the coding exons specific to *lin-29a*, at the same time introducing a frameshift in the downstream *lin-29a* reading frame. By injecting two sgRNAs (sgRNA1: gctggaac-caccactggctc, sgRNA2: gtggcaggagagaattctga), we obtained the *lin-29a(xe40) = lin-29(Δa)* allele, a 1,102-bp deletion covering exons 2–4, introducing a frameshift in the *lin-29a* reading frame with a predicted stop codon in exon 6. The deletion has the following flanking sequences: 5′ ctctggaatagctggaaccac—*xe40* deletion—attctctcctgccacatcat 3′. Translation of *xe40* yields a protein with the N-terminal 22 amino acids of LIN-29A (MDQTVLDSAFNSPVDSGIAGTT), followed by a stretch of 69 out-of-frame amino acids and a stop codon.

### Isoform-specific GFP::3xFLAG tagging of endogenous *lin-29a* using CRISPR-Cas9

To specifically tag LIN-29a at the N terminus, the following mix was injected into wild-type worms (Dickinson et al, 2015; Katic et al, 2015): 50 ng/μl pIK155, 100 ng/μl of pIK198 with a cloned sgRNA template (atattatttatcagtgattg), 2.5 ng/μl pCFJ90, 5 ng/μl pCFJ104, and 10 ng/μl pDD282 with cloned homology arms. Recombinants were isolated according to the protocol by Dickinson et al (2015), verified by DNA sequencing and outcrossed three times. The plasmid for homologous recombination, pFA27, was prepared by restriction digest of pDD282 with ClaI and SpeI, followed by a Gibson assembly reaction (Gibson et al, 2009) with two gBlocks Gene Fragments (Integrated DNA Technologies) (Table S6).

### Generation of a balancer allele for *lin-41(xe8)* using CRISPR-Cas9

The *lin-41(xe8) = lin-41(ΔLCS)* allele is not temperature sensitive like *let-7(n2853)* and, therefore, causes lethality at any temperature (Ecsedi et al, 2015). To maintain *lin-41(xe8)* animals, a balancer null allele, *lin-41(bch28)*, was previously created by inserting an expression cassette driving ubiquitous nuclear GFP from the *eft-3* promoter into the *lin-41* coding sequence (Katic et al, 2015). To avoid generating a wild-type *lin-41* copy—and a recombined *lin-41(bch28 xe8)* allele—through intragenic recombination of *lin-41(bch28)* with *lin-41(xe8)*, we additionally deleted a large part of the *lin-41* coding sequence together with the part of the *lin-41* 3′ UTR containing the *let-7* complementary sites within the *lin-41(bch28)* allele. To this end, *lin-41(bch28)* heterozygous worms were injected with a mix containing 50 ng/μl pIK155, 100 ng/μl of each pIK198 with a cloned sgRNA, 5 ng/μl pCFJ90, and 5 ng/μl pCFJ104, as previously described (Katic et al, 2015). We injected two plasmids encoding sgRNAs, sgRNA1 (ggtgactgaatcattgacgg) and sgRNA2 (agaaggtttcaatggttcag), cutting in the third coding

exon and the 3′ UTR of *lin-41*, respectively. Single F1 progeny of injected wild-type worms were picked to individual plates and the F2 progeny were screened for expected deletions in *lin-41(bch28)* by PCR. The *lin-41(bch28 xe70)* allele thus obtained was further validated by DNA sequencing and outcrossed three times to the wild-type strain before crossing it with *lin-41(xe8)* heterozygous animals. The final *lin-41(bch28 xe70)* allele consists of the inserted expression cassette, as described in Katic et al (2015), followed by an additional deletion of the region with the following flanking sequences: 5′ ggctcactatttgacactcc—*xe70* deletion (6,395 bp)—accattgaaaccttctccc 3′.

### Construction of transgenic single-copy GFP reporters

Transgenes were cloned into the destination vector pCFJ150 (Frøkjaer-Jensen et al, 2008) using the MultiSite Gateway Technology (Thermo Fisher Scientific) as described in Table S6. Worm lines with integrated transgenes were obtained by Mos1-mediated single-copy integration (MosSCI) into chromosome II (*ttTi5605* locus), using a protocol for injection with low DNA concentration (Frøkjær-Jensen et al, 2012).

### Microscopy

For confocal imaging of the *mab-3* and *dmd-3* reporter worm lines (HW1803, HW1798, HW1827, and HW1828), synchronized arrested L1 stage larvae were grown for 20 h at 25°C on RNAi-inducing plates with HT115 bacteria. The bacteria either contained the L4440 parental RNAi vector without insert (denoted "mock RNAi") or with an insert targeting *lin-41* (Fraser et al, 2000). For confocal imaging of endogenously tagged GFP::LIN-29a (HW1826 and HW1882), worms were grown at 25°C on OP50 bacteria. Worms were imaged on a Zeiss LSM 700 confocal microscope driven by Zen (2012) Software after mounting them on a 2% (wt/vol) agarose pad with a drop of 10 mM levamisole solution. For imaging of HW2295 (*xeSi417* transgene expression), a Zeiss Axio Observer Z1 microscope was used. Differential Interference Contrast and fluorescent images were acquired with a 40×/1.3 oil immersion objective (1,024 × 1,024 pixels, pixel size 156 nm). Using the Fiji software (Schindelin et al, 2012), the images were processed after selecting representative regions. Worms of the same worm line were imaged and processed with identical settings.

### Phenotype quantification

To image or quantify phenotypes, arrested L1 larvae were grown at 25°C until the synchronized population reached the desired developmental stage. For worm lines containing mnC1-balanced animals, balanced animals were identified by *myo-2p::gfp* expression and excluded from the analysis. Similarly, to score *lin-41(xe8)* homozygous animals within a population of balanced *lin-41(xe8)/lin-41(bch28 xe70)* animals, all *eft-3p::gfp::h2b* expressing (i.e., balancer carrying) animals were excluded from the analysis. Images were acquired on a Zeiss Axio Observer Z1 microscope using the AxioVision SE64 software. Animals were mounted on a 2% (wt/vol) agarose pad and immobilized in 10 mM levamisole. Selection of

regions and processing of images was performed with Fiji (Schindelin et al, 2012).

To quantify seam cell numbers, we counted all clearly visible fluorescent cells of the upper lateral side in mounted worms expressing a seam cell–specific *scm::gfp* transgene (Koh & Rothman, 2001). Synchronized worms were grown at 25°C for 36–38 h (late L4 stage) or 40–42 h (young adult stage), with the exact developmental time assessed by staging of individual worms according to gonad length and vulva morphology.

To examine male tail retraction defects, we used *him-5 (e1490)*–mutant worms in different genetic backgrounds. Images were acquired after growing males to early, mid, or late L4 larvae as well as to young adults. To quantify tail retraction defects, male tail phenotypes were scored in mounted worms at the late L4 and at the young adult stage. At the late L4 stage, tail retraction defects were categorized as unretracted (no cell retraction visible), partially retracted (cells less retracted than in wild-type) or over-retracted tails (cells more retracted than in wild-type). At the young adult stage, abnormal tails were categorized into over-retracted, Lep, and unretracted (very long spike, compromised rays and fan structures) phenotypes. Tails with a spike extending beyond the fan were counted as "Lep" and tails with smaller spikes were considered wild-type (as in our hands, those were difficult to distinguish from wild-type tails). Whereas most *lin-41(bx37)*– or *lin-41(bx42)*–mutant males had a Lep tail, all *lin-41(xe8)* animals showed the distinct and more severe "unretracted" phenotype. 85% of *let-7(n2853)*–mutant males also displayed a completely unretracted tail. In 15% of the scored *let-7(n2853)*–mutant males, the tail cell started retracting at the time point of analysis (young adult stage), still resulting in a similarly elongated tail, but with a rounded tip. This phenotype was categorized as "unretracted," as it was more similar to fully unretracted tails than to Lep tails. The observed delayed start of the retraction program may be due to residual *let-7* activity in the *let-7 (n2853)ts* mutant.

To examine vulval phenotypes (bursting, Pvl, and Egl), worm phenotypes were scored directly on the nematode growth medium agar plates using a dissecting microscope. Vulval bursting and Pvl phenotypes were quantified in synchronized worm populations grown at 25°C for 45 h; a time ~5 h after the first *let-7(n2853)* or *lin41 (xe8)* animals burst through their vulva. Animals with part of the intestine protruding through the vulva were counted as "burst," all other animals as "non-burst." Pvl phenotypes were only scored in non-burst animals. Egl phenotypes were scored for non-burst worms in synchronized populations grown at 25°C for 60 h. At least 400 worms were scored for each genotype in each of these experiments to quantify vulval phenotypes.

To quantify uterine-vulval connection phenotypes, worms were grown in synchronized populations at 25°C for 36–38 h to reach the late L4 larval stage. Phenotypes of mounted worms were assigned to one of two categories, worms with a normal thin utse structure and worms with an abnormal thicker cell layer, and counted.

### RIP-seq

RNA co-immunoprecipitations were performed with semi-synchronous L3/L4 stage populations of *lin-41(n2914); him-5 (e1490)*–mutant worms expressing *flag::gfp::lin-41* (Aeschimann et al,

2017) or of *him-5(e1490)*–mutant worms expressing *flag::gfp::sart-3* (Rüegger et al, 2015) as previously described (Aeschimann et al, 2017). RNA bound to the beads and input RNA was extracted using Tri Reagent (Molecular Research Center; TR 118) according to the manufacturer's recommendations. The input RNA samples were diluted to match the low RNA concentrations of 5–10 ng/$\mu$l of the immunoprecipitated RNA. Sequencing libraries were prepared with the TruSeq Stranded mRNA HT Sample Prep kit (RS-122-2103; Illumina) and 50-bp single end reads were sequenced on an Illumina HiSeq2000 machine.

PCR duplicates were first removed by collapsing reads with an identical 5′ end coordinate to a single read. De-duplicated reads were then aligned to the May 2008 (ce6) *C. elegans* genome assembly from University of California, Santa Cruz (Rosenbloom et al, 2015). Alignments were performed using the qAlign function from the QuasR R package (v. 1.20.0) (Gaidatzis et al, 2015), with the reference genome package ("BSgenome.Celegans.UCSC.ce6") downloaded from Bioconductor (https://www.bioconductor.org) and with the parameter "splicedAlignment=TRUE," which calls the SpliceMap aligner with default parameters (Au et al, 2010). The resulting alignments were converted to BAM format, sorted, and indexed using SAMtools (version 1.2) (Li et al, 2009). Gene coverage was quantified using annotations downloaded from WormBase (version WS190; ftp://ftp.wormbase.org/pub/wormbase/releases/WS190/). Reads overlapping all annotated exons for each gene were counted. For plotting, samples were normalized by the mean number of counts mapping to exons in all samples and then log$_2$-transformed after adding a pseudocount of 8. To determine genes significantly bound by LIN-41, a model was constructed using edgeR (v. 3.22.3) (Robinson et al, 2010) containing a term for sequencing batch, library type (IP or input), and protein (LIN-41 or SART-3), as well as an interaction term between library type and protein. Testing for significance of the interaction term with a likelihood ratio test identified genes for which the enrichment of IP versus input was significantly greater for LIN-41 than for the SART-3 control. After conducting a multiple hypothesis test correction, we applied a cutoff of FDR < 0.05 to determine a final set of bound genes. All computations were performed using R (v. 3.5.1).

### Data availability

All RIP-seq data generated in this study have been deposited in the National Center for Biotechnology Information Gene Expression Omnibus (GEO) (Edgar et al, 2002) under GEO Series accession number GSE120405.

# Supplementary Information

# Acknowledgements

We thank Iskra Katic and Jun Liu for generating the *lin-41(bch28 xe70)* balancer allele, Sarah Carl for help with computational analysis of RIP-

Seq data, Monika Fasler and Lan Xu for help with generating strains, and Benjamin Towbin, Oliver Hobert, Laura Pereira, Iskra Katic, Chiara Azzi, and Jun Liu for critical feedback on the manuscript. We thank Adam Mason and Douglas Portman for providing strains. Some strains were provided by the Caenorhabditis Genetics Center, which is funded by the NIH Office of Research Infrastructure Programs (P40 OD010440). This project has received funding from the Swiss National Science Foundation (#31003A_163447), the European Research Council under the European Union's Horizon 2020 research and innovation program (grant agreement no. 741269), and Friedrich Miescher Institute for Biomedical Research core funding through the Novartis Research Foundation (to H Großhans).

## Author Contributions

F Aeschimann: conceptualization, formal analysis, supervision, investigation, visualization, methodology, and writing—original draft, review, and editing.
A Neagu: formal analysis and investigation.
M Rausch: resources and investigation.
H Großhans: conceptualization, supervision, funding acquisition, project administration, and writing—original draft, review, and editing.

## Conflict of Interest Statement

The authors declare that they have no conflict of interest.

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
