## [Reviewer comments · Life Science Alliance]

Life Science Alliance

let-7 coordinates the transition to adulthood through a single primary and four secondary targets

Florian Aeschimann, Anca Neagu, Magdalene Rausch, and Helge Grosshans
DOI: <https://doi.org/10.26508/lsa.201900335>

Corresponding author(s): Helge Grosshans, Friedrich Miescher Institute for Biomedical Research

Review Timeline:

Submission Date:	2019-02-08
Editorial Decision:	2019-02-08
Revision Received:	2019-03-04
Editorial Decision:	2019-03-04
Revision Received:	2019-03-06
Accepted:	2019-03-06

Scientific Editor: Andrea Leibfried

Transaction Report:

Please note that the manuscript was previously reviewed at another journal and the reports were taken into account in inviting a revision for publication at *Life Science Alliance* prior to submission to *Life Science Alliance*.

February 8, 2019

Re: Life Science Alliance manuscript #LSA-2019-00335

Helge Grosshans
Friedrich Miescher Institute for Biomedical Research
Maulbeerstrasse 66
Basel 4058
Switzerland

Dear Dr. Grosshans,

Thank you for transferring your manuscript entitled "let-7 controls the transition to adulthood by releasing select transcriptional regulators from repression by LIN41" to Life Science Alliance. The manuscript was assessed by expert reviewers at another journal before, and the editors transferred those reports to us with your permission.

The reviewers appreciated that your work confirms and extends previous conclusions on the role of LIN41 with more sophisticated tools and that you demonstrate redundant action of LIN-29a and MAB-10. Based on the reports already at hand, we would therefore like to invite you to provide a revised version of your work for publication in Life Science Alliance. We would expect a point-by-point response to reviewer #2's concerns and accordingly text changes/further discussion and that you show the effect of the lin-41 allele on its own (point 1) as well as the expression pattern of the ectopic LIN-29::GFP (point 3).

Thank you for this interesting contribution to Life Science Alliance. We are looking forward to receiving your revised manuscript.

Sincerely,

Andrea Leibfried, PhD
Executive Editor
Life Science Alliance
Meyerhofstr. 1
69117 Heidelberg, Germany
t +49 6221 8891 502

- A letter addressing the reviewers' comments point by point.
- An editable version of the final text (.DOC or .DOCX) is needed for copyediting (no PDFs).
- High-resolution figure, supplementary figure and video files uploaded as individual files: See our detailed guidelines for preparing your production-ready images, <http://life-science-alliance.org/authorguide>
- Summary blurb (enter in submission system): A short text summarizing in a single sentence the study (max. 200 characters including spaces). This text is used in conjunction with the titles of papers, hence should be informative and complementary to the title and running title. It should describe the context and significance of the findings for a general readership; it should be written in the present tense and refer to the work in the third person. Author names should not be mentioned.

B. MANUSCRIPT ORGANIZATION AND FORMATTING:

Full guidelines are available on our Instructions for Authors page, <http://life-science-alliance.org/authorguide>

March 4, 2019

RE: Life Science Alliance Manuscript #LSA-2019-00335RR

Dr. Helge Grosshans
Friedrich Miescher Institute for Biomedical Research
Maulbeerstrasse 66
Basel 4058
Switzerland

Dear Dr. Grosshans,

Thank you for submitting your revised manuscript entitled "let-7 coordinates the transition to adulthood through a single primary and four secondary targets". I appreciate the introduced changes and am thus happy to accept your manuscript in principle for publication here.

Before sending you the final acceptance letter, please log into our system once more to fill in the electronic license to publish form. Please make sure to move all manuscript files to the submission version with the license (single click process). Please have the first author check whether the ORCID iD associated with his profile is correct - the difficulties you/he encountered were due to the fact that our system had several profiles with the same name, one of which was already linked to an ORCID iD.

A. FINAL FILES:

-- Summary blurb (enter in submission system): A short text summarizing in a single sentence the study (max. 200 characters including spaces). This text is used in conjunction with the titles of papers, hence should be informative and complementary to the title. It should describe the context

and significance of the findings for a general readership; it should be written in the present tense and refer to the work in the third person. Author names should not be mentioned.

B. MANUSCRIPT ORGANIZATION AND FORMATTING:

Sincerely,

March 6, 2019

RE: Life Science Alliance Manuscript #LSA-2019-00335RRR

Dr. Helge Grosshans
Friedrich Miescher Institute for Biomedical Research
Maulbeerstrasse 66
Basel 4058
Switzerland

Dear Dr. Grosshans,

Thank you for submitting your Research Article entitled "let-7 coordinates the transition to adulthood through a single primary and four secondary targets". It is a pleasure to let you know that your manuscript is now accepted for publication in Life Science Alliance. Congratulations on this interesting work.

DISTRIBUTION OF MATERIALS:

Again, congratulations on a very nice paper. I hope you found the review process to be constructive and are pleased with how the manuscript was handled editorially. We look forward to future exciting submissions from your lab.

Sincerely,
